# Event Tokenization and Next-Token Prediction for Anomaly Detection at the Large Hadron Collider

**Ambre Visive[1,2]⋆, Polina Moskvitina [3,2], Clara Nellist [1,2], Roberto Ruiz de Austri[4] and Sascha Caron[3,2]**

**1** Institute of Physics, University of Amsterdam, Amsterdam,The Netherlands
**2** Nikhef, Dutch National Institute for Subatomic Physics, Amsterdam, The Netherlands
**3** High Energy Physics, Radboud University, Nijmegen, The Netherlands
**4** Instituto de Física Corpuscular, IFIC-UV/CSIC, Paterna, Spain

⋆ ambre.visive@cern.ch

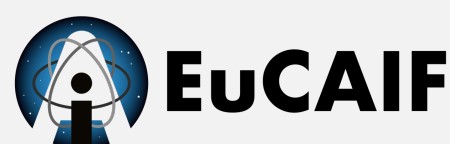

*The 2nd European AI for Fundamental Physics Conference (EuCAIFCon2025) Cagliari, Sardinia, 16-20 June 2025*

## Abstract

We propose a novel use of Large Language Models (LLMs) as unsupervised anomaly detectors in particle physics. Using lightweight LLM-like networks with encoder-based architectures trained to reconstruct background events via masked-token prediction, our method identifies anomalies through deviations in reconstruction performance, without prior knowledge of signal characteristics. Applied to searches for simultaneous four-top-quark production, this token-based approach shows competitive performance against established unsupervised methods and effectively captures subtle discrepancies in collider data, suggesting a promising direction for model-independent searches for new physics.

## 1 Introduction

Large Language Models (LLMs), originally developed for natural language tasks [1], have shown remarkable capabilities in modeling complex data distributions [2]. Their success is largely driven by transformer architectures [3] and large-scale training on diverse datasets [4]. In high-energy physics, the increasing data volume from the LHC opens new opportunities for applying such models [5]. In this work, we explore LLM-like networks as potential unsupervised anomaly detection techniques, trained to reconstruct background events and identify deviations in the reconstruction without prior signal knowledge. This approach aims to improve sensitivity to rare Standard Model processes and uncover potential Beyond the Standard Model signatures.

## 2 Dataset and Model

### 2.1 Physics Motivation

Four-top-quark events present a complex final state, with 0-4 leptons and 4-12 jets, including four from bottom quarks, due to the dominant decay channel $t \rightarrow W + b$, as $|V_{tb}|^2 \sim 1$. Their signature closely resembles that of $t\bar{t}WW$ and $t\bar{t}W$, differing mainly by additional $b$-jets. Similarly, $t\bar{t}Z$, and $t\bar{t}H$ processes can be confused with four-top-quark signatures, as $Z$ and Higgs bosons often decay into jets or leptons. Due to these overlaps, we treat $t\bar{t}WW$, $t\bar{t}W$, $t\bar{t}Z$ and $t\bar{t}H$ as **background events** in this work, while $t\bar{t}t\bar{t}$ is the **signal process**, we are trying to isolate. Its rarity and complex topology make it an ideal benchmark for evaluating unsupervised anomaly detection methods on Standard Model (SM) processes.

### 2.2 The Dark Machines Datasets and Data Format

This study uses SM datasets from the Dark Machines challenge [6], comprising over $10^9$ simulated $pp$ collisions at $\sqrt{s} = 13$ TeV. Events are generated with `MG5_aMC@NLO` 2.7, hadronized using `Pythia` 8.239, and processed through the `Delphes` 3.4.2 with modified ATLAS-Run 2 settings. The generated Monte-Carlo data, stored in ROOT format, were converted to CSV files following the event selection in [7]. In a second pre-processing step, the event-type is extracted and mapped to an integer as in Table 1 to create labeled datasets. Each event is represented as a sequence of particle-object tags (see Table 2) and their four-momenta (energy $E$, transverse momentum $p_T$, pseudo-rapidity $\eta$, azimuthal angle $\phi$), ordered by type and $p_T$, and the missing transverse energy of the event ($\|E_T^{miss}\|$) and its azimuthal angle ($\phi_{E_T^{miss}}$). Bottom-quark jets are tagged without uncertainty. Sequences are padded to a fixed length. The dataset is then split into training (80%), validation (10%), and test (10%) subsets. Further details are available in [6–8].

$$obj_1; obj_2; ...; obj_{18}; \|E_T^{miss}\|; \phi_{E_T^{miss}}; E_1; p_{T,1}, \eta_1; \phi_1; E_2; p_{T,2}, \eta_2; \phi_2; ...; E_{18}; p_{T,18}, \eta_{18}; \phi_{18}$$

The input to the LLM-like model consists of batches of token sequences, each representing a particle physics event. Tokens encode the missing transverse energy, its azimuthal angle, or the particle, the latter being characterised by its type, charge, transverse momentum, pseudo-rapidity, and azimuthal angle. Constructing these sequences from the dataset requires a dedicated tokenization step.

| SM process | $t\bar{t}t\bar{t}$ | $t\bar{t}H$ | $t\bar{t}W$ | $t\bar{t}WW$ | $t\bar{t}Z$ |
|---|---|---|---|---|---|
| process ID | 1 | 2 | 3 | 4 | 5 |

Table 1: Labels of SM processes.

| Object | jet | b-tagged jet | positron | electron | muon | anti-muon | photon |
|---|---|---|---|---|---|---|---|
| symbol ID | j | b | e+ | e- | mu+ | mu- | g |
| tag | 1 | 2 | 3 | 4 | 5 | 6 | 7 |

Table 2: Tags of particle-objects.

### 2.3 Large-Language-Model for Anomaly Detection and Tokenization Strategy

We employ a lightweight LLM-like model based on the transformer encoder. Input sequences, tokenized representations of particle physics events, are embedded. The core of the model consists of two transformer layers with four self-attention heads each, enabling the model to capture contextual relationships across tokens. It is followed by a linear projection and

softmax layer that outputs a probability distribution over token classes, indicating the likelihood of each token being the correct reconstruction. To have the model learn the distribution of background events, training is performed on background events only, using masked-token prediction: one token per event is randomly masked, and the model is trained to reconstruct it using Sparse Categorical Cross-Entropy loss. Optimization is done with Adam [9], and early stopping is applied. During inference, all tokens in each event are masked and reconstructed one at a time. The reconstruction scores are averaged to produce a reconstruction score per event. Events with poor reconstruction are flagged as anomalous, indicating deviation from the learned background distribution. This yields score distributions (see Figure 1), from which thresholds can be defined to identify potential signal events.

A tokenization step is needed to represent particle physics events as sequences of tokens suitable for LLM or LLM-like models. The tokens are obtained through a binning strategy: while particles are mapped to seven predefined categories (see Table 2), different discretization intervals for $p_T$, $\eta$, and $\phi$ were tested as careful selection of bin edges ensures meaningful separation of physical features. Performances of each binning-based tokenization strategy and the impact of including $\|E_T^{miss}\|$ and $\phi_{E_T^{miss}}$ as additional tokens in the sequence representation were assessed using a compact classifier neural network or the downstream LLM-like model performance. In the most effective binning configuration:

- $p_T$, $\eta$ and $\|E_T^{miss}\|$ were each divided into 4 bins, with edges defined as containing 25% of the background data in each bin.

- $\phi$ and $\phi_{E_T^{miss}}$ were divided into 4 bins of width $\frac{1}{4}\pi$.

- With bins indexed from 1 onward, each particle token is defined as followed:
  $token_{part} = 64 \times (bin_{obj} - 1) + 16 \times (bin_{p_T} - 1) + 4 \times (bin_{\eta} - 1) + bin_{\phi}$.

- It yields: $\cdot\, token_{part} \in [1, 448]$;   $\cdot\, token_{\|E_T^{miss}\|} \in [449, 452]$;   $\cdot\, token_{\phi_{E_T^{miss}}} \in [453, 456]$.

- An event is represented as a sequence:
  $[token_{part,1}, token_{part,2}, token_{part,3}, ..., token_{part,18}, token_{\|E_T^{miss}\|}, token_{\phi_{E_T^{miss}}}]$

Since the original dataset is zero-padded and transformer models require uniform sequence lengths across batches, $token_{part,i} = 0$ was set for padded entries where no particle is present.

This tokenization enables the LLM-like model to effectively learn the structure of background events. However, it is possible that a deep-learned tokenization would yield better results.

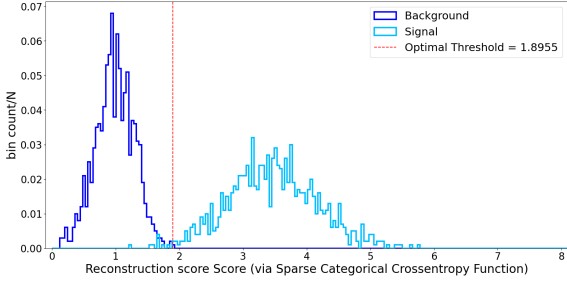

Figure 1: Illustrative distribution of the aggregated reconstruction scores in a perfectly trained model, evaluated with sparse categorical cross-entropy, for background (blue) and signal (cyan) events. The red dashed line indicates the optimal threshold used to separate the two classes.

# 3 Results and Evaluation

## 3.1 Search for Four-Top Production

To evaluate the performance of the method on the simultaneous four-top-quarks production, both the background and the signal are tokenized using the strategy described in Subsection 2.3. Once the LLM-like model has been trained on the background data, both signal and background data are injected. Following the technique described in 2.3, the distribution of the average reconstruction score of the signal and background events is obtained, as shown in Figure 2a. The common area of 70.85% illustrates the overlap between the distributions, highlighting the model's ability to discriminate between background and signal events. The optimal threshold used to separate the two classes can be yielded and used as reference for future analysis. From this plot, the Receiving Operator Characteristic (ROC) curve can be derived (see Figure 2b) and the area under the curve (ROC-AUC) of the model can be calculated. The model yields: ROC-AUC = 0.67.

## 3.2 Comparison with Established Unsupervised Methods

A comparison was conducted with established unsupervised anomaly detection methods implemented in [7]: the DDD, DeepSVDD and DROCC methods. As illustrated in Figure 2b, although the proposed method does not surpass the DDD-based techniques, improved performance was observed relative to DeepSVDD and DROCC, indicating competitive behavior in unsupervised techniques for anomaly detection.

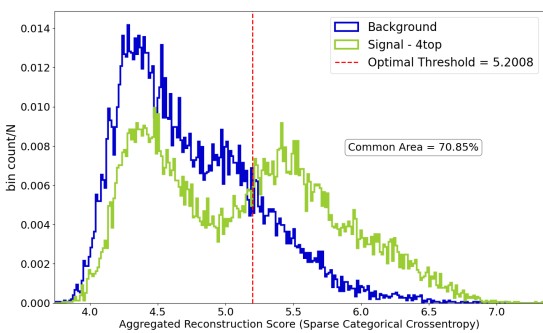

(a) Distribution of the aggregated reconstruction scores, evaluated with sparse categorical cross-entropy, for background (blue) and four-top-signal (green) events. The red dashed line indicates the optimal threshold used to separate the two classes.

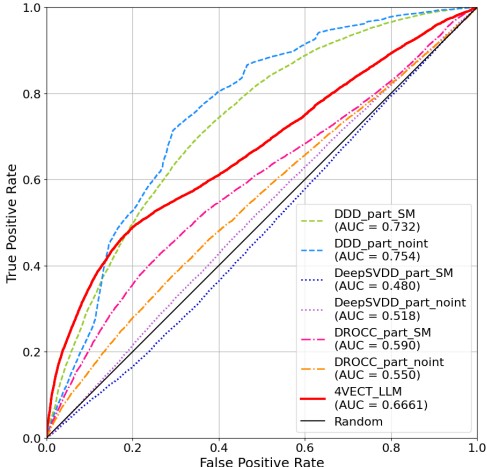

(b) Comparison of the ROC curves of different anomaly detection methods

Figure 2: Results from the LLM-like method and comparisons to alternative methods.

# 4 Conclusion

A novel application of LLM-like models for unsupervised anomaly detection in particle physics has been presented. Although the results remain preliminary, promising performance was observed in identifying rare processes such as four-top-quark production. While not outperforming all existing approaches, competitive results were achieved, supported by a flexible token-based representation of collider data. With further optimization of the tokenization scheme and model architecture, improvements in sensitivity and robustness are anticipated, making this approach a viable candidate for model-independent searches in future high-energy physics analyses.

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
