# Peer review of "Event Tokenization and Next-Token Prediction for Anomaly Detection at the Large Hadron Collider"

_SciPost Physics Proceedings_

## Round 1 · Referee Report · Anonymous (Referee 1) · 2025-11-6

Strengths

  1. The physics motivation is clear
  2. The data is well described
  3. The idea is clearly described and motivated

Weaknesses

None, other than the length of the paper but that is obviously not something the authors can change.

Report

The article meets the Journal's acceptance criteria.

This work uses the reconstruction loss of a transformer trained on a masked token prediction task to derive anomaly scores. The model is trained on background only, and the idea is that the reconstruction loss for background-like events should be low, whereas for unseen data (like a BSM signal) it should be high. In order to be able to use cross-entropy loss, the input data must be tokenized. This is done with binning.

Requested changes

  1. To the best of my knowledge, this is the first work using reconstruction loss from a masked token prediction task for unsupervised anomaly detection in high-energy physics. However, I wonder if there are examples from other fields (example: LAnoBERT), and would recommend including some references to previous work of that type.

  2. I question the use of the term "LLM-like model", as this is simply a transformer encoder trained on tokenized data, which in itself does not have anything to do with LLMs. LLMs may also use transformers and tokenized data, but they are also both finetuned and generative, which this model is not. I suggest to drop the use of this term, apart from in the beginning possibly when referring to BERT where the masked token training was introduced -- please add this reference. Fig 2b refers to the model as 4VECT_LLM, and the conference poster refers to it simply as 4VECT, which makes me think that this is the name of the model although it is never introduced as such in the text.

  3. Page 2, middle, paragraph starting with "The input to the LLM-like model...": Since this talks about tokenization, this fits better in section 2.3. At its current position it is a bit confusing to the reader.

  4. What is the max padding length of the sequences? On page 2 it looks like at most 18 particles per event are stored, is this the case? If yes, please specify.

  5. Page 3, middle paragraph: what is this compact classifier neural network that was used to evaluate the performance of the binning strategy?

  6. All figures: please make the text and tick labels larger, this is very small

  7. Check capitalization in the titles in references (eg. "lhc")

Recommendation

Ask for minor revision

---

## Editorial Decision

awaiting_resubmission